# "If I am alive, I am happy": Defining quality of care from the perspectives of key maternal and newborn health stakeholders in Papua New Guinea

**Lachlan M. Faktor**[1,2]*, **Alyce N. Wilson**[1], **Pele Melepia**[3], **Delly Babona**[1,4], **Pinip Wapi**[5], **Rose Suruka**[3], **Priscah Hezeri**[3], **Duk Duk Kabiu**[3], **Lisa M. Vallely**[6,7], **Elissa Kennedy**[1], **Michelle J. L. Scoullar**[1], **Naomi Spotswood**[1,8,9], **Caroline S. E. Homer**[1]

1 Maternal, Child and Adolescent Health Program, International Development, Burnet Institute, Melbourne, Australia, 2 School of Public Health and Preventative Medicine, Monash University, Melbourne, Australia, 3 Healthy Mothers, Healthy Babies Program, Burnet Institute, Kokopo, Papua New Guinea, 4 St Mary's Hospital Vunapope, Kokopo, East New Britain Province, Papua New Guinea, 5 Nonga General Hospital, Rabaul, Papua New Guinea, 6 Papua New Guinea Institute for Medical Research, Goroka, Papua New Guinea, 7 Kirby Institute, University of New South Wales, Kensington, Australia, 8 Royal Hobart Hospital, Hobart, Australia, 9 Department of Medicine, University of Melbourne, Melbourne, Australia

* lachlan.faktor@student.burnet.edu.au

**Data Availability Statement:** The datasets generated and/or analysed during the current study are not publicly available due to potential confidentiality concerns. Additional information can

## Abstract

Quality maternal and newborn healthcare is essential to improve experiences and health outcomes for mothers and babies. In many low to middle income countries, such as Papua New Guinea, there are initiatives to increase antenatal care attendance and facility births. To develop and implement initiatives that are appropriate, relevant, and contextualised to a community, it is important to understand how quality of care is perceived and defined by different maternal and newborn healthcare stakeholders. The aim of this study was to understand how women, their partners, healthcare professionals, healthcare managers, and provincial health administrators in East New Britain, Papua New Guinea define quality of pregnancy, childbirth, and immediate postnatal care. An exploratory qualitative study underpinned by a partnership-defined quality approach was undertaken. In total, 42 participants from five different healthcare facilities in East New Britain, Papua New Guinea, were interviewed. These included women, partners, healthcare professionals, healthcare managers, and provincial health administrators. Interviews were analysed using thematic analysis, assisted by NVivo computer software. Four themes were identified aligning with the journey a woman takes throughout the health system. These included (I) Ensuring Access: Arriving at the health centre, (II) Experiencing Positive Care: What the staff do, (III) Having the Bare Minimum: Resources available to the service, and (IV) Meeting Expectations: Outcomes of care. Stakeholder groups had significant overlap in how quality of care was defined, however women and partners focussed more on elements relating to experience of care, while clinical stakeholders focussed on elements relating to provision of care. There is a gap in how stakeholders define quality maternal and newborn healthcare, and the quality of the care which is administered and received.

be made available from the Scientific Integrity Officer at Burnet Institute, (admin@burnet.edu.au), on reasonable request. These restrictions are per the Papua New Guinea, Institute of Medical Research, Institutional Review Board.

**Funding:** Funding was provided by the Burnet Institute through philanthropic support from numerous private and business donors in Australia and PNG. Major funding was provided by June Canavan Foundation Australia; Gras Foundation, Australia; Bank South Pacific PNG Community Grant; Steamships PNG Community Grant; Alistair Lucas Prize for Medical Research; National Health and Medical Research Council (NHMRC) of Australia (Fellowships to CSE and JGB, Postgraduate Research Scholarship to AW); Naylor Steward Ancillary Fund, and the Chrysalis Foundation. MJLS received a Basser Research Entry Scholarship from the Royal Australasian College of Physicians Foundation (2018 and 2020). Burnet Institute is supported by an Operational Infrastructure Grant from the State Government of Victoria, Australia, and the NHMRC Independent Research Institutes Infrastructure Support Scheme. The funders had no role in the study design, data collection and analysis, decision to publish, or preparation of the manuscript.

**Competing interests:** The authors have declared that no competing interests exist.

## Introduction

Papua New Guinea (PNG) is the largest of the small island nations of the Pacific region, with a population estimated at over ten million [1]. Multiple barriers to accessing maternal and newborn health care exist including a predominantly rural population (86.8%) [2], difficult geography including mountainous terrain, as well as numerous hard-to-access islands. Chronic health workforce shortages [3], limited infrastructure and consumables, high rates of adolescent pregnancy and premature birth are additional challenges to achieving high quality care [4–6]. While estimates of the Maternal Mortality Ratio in PNG vary (from 215 to 930 per 100,000 live births), the broadly accepted figure is around 500 maternal deaths per 100,000 live births, one of the highest in the Pacific region [7]. The Neonatal Mortality Ratio (20 per 1,000 live births) and stillbirth rate (20–30 per 1,000 live births) are similarly high [8]. These outcomes are in part due to low rates of access to care throughout the pregnancy and birth journey. Across PNG, only 48% of pregnant women (S1 Text) attended one or more antenatal care visits and 36% gave birth in a facility in 2020 [9].

There is increasing recognition that high quality care leads to optimum health outcomes [10, 11]. The World Health Organization's definition of quality care is care which is effective, safe, people-centred, timely, equitable, integrated and efficient [12], and is one of several accepted definitions [13, 14], along with specific definitions focussed on quality of maternal and newborn care [15–17]. However, these definitions do not necessarily capture how key stakeholders, the providers, and receivers of care, see, define, or experience quality of care. Understanding quality of care from different perspectives can inform interventions, engage users as active participants in service delivery, and encourage community collaboration in co-designing and implementing quality improvement interventions and initiatives [18]. Key maternal and newborn health stakeholders include clients of the services, their partners and families, healthcare professionals (including midwives, nurses, obstetricians, and paediatricians), traditional birth attendants, community members, health managers, and health administrators. In PNG, the maternity workforce also includes community health workers and health extension officers; specially trained health workers who assist with maternity care of women, daily administration, and coordination of community health services [19].

Previous research across low and middle income countries (LMICs) has found that definitions of quality of care from both women and providers of maternal and newborn health care include aspects such as timeliness of care, autonomy, adequate human and physical resources, respectful care, and privacy, and there can be similarity between stakeholder groups [20–23]. While there can be similarities in quality of care priorities between stakeholder groups, women tend to have a greater focus on the care experience, such as positive relationships with caregivers and woman-centred care, while providers of care tend to emphasise structural elements relating to the provision of care, such as availability of resources [24, 25]. Evidence from the Asia Pacific region defining perspectives of quality maternal and newborn health care is limited, with much of the literature in LMICs coming from Sub-Saharan Africa. To our knowledge, there are no such published reports from PNG. The aim of this study was to inform this knowledge gap and describe how women, their partners, healthcare professionals, healthcare managers, and provincial health administrators in East New Britain, PNG define quality of pregnancy, childbirth, and immediate postnatal care.

## Methods

### Ethics statement

This study received ethical approval in PNG from the PNG Institute of Medical Research Institutional Review Board (1903) and the National Department of Health Medical Research

Advisory Committee (19.16), and in Australia from the Alfred Hospital Human Research Ethics Committee (267/19). All participants involved in the study provided written and verbal informed consent.

## Study design and setting

This study used an exploratory, qualitative methodology, and a phenomenological approach which aimed to understand the lived experiences and perspectives of maternal and newborn health care stakeholders in East New Britain. This study was co-designed by the Burnet Institute and the East New Britain Provincial Health Authority, using a partnership-defined quality approach to integrate community involvement and mobilisation [26]. This facilitated the project being driven by local preferences and needs. This study is couched within the overarching Healthy Mothers, Healthy Babies program, and is known as the Gutpela Helt Sevis, Helti Mama Bel, Helti Beibi Stadi (Quality of Pregnancy, Childbirth and Newborn Health Services Study), which aims to identify low-cost, effective, and feasible quality improvement interventions. Qualitative methods allowed researchers to capture the complexity of different stakeholder definitions of quality maternal and newborn care, and explore the experience associated with that care [27]. Findings are reported according to the consolidated criteria for reporting qualitative research (COREQ) checklist (S1 Table) [28]. This study took place in the rural island province of East New Britain, which has a population of around 400,000 [9]. In addition to English and Tok Pisin, multiple local languages are spoken by the three main cultural groups–the Baining, Pomio and Tolai people. Whilst there is access via roads to larger towns and villages, many rural and coastal communities are only accessible via walking track and/or boat [9]. As in wider PNG, East New Britain suffers from a chronic health workforce shortage, with 15 healthcare workers per 10,000 [9], around a third of the WHO recommended 44.5 per 10,000 [29]. East New Britain has one tertiary health service (Nonga General Hospital), three rural hospitals, 32 health centres, and 109 community health posts. The Provincial Health Authority are ultimately responsible for these centres, however Catholic and other faith-based services manage approximately 50 per cent of health facilities. The National Department of Health expects that all facilities at all levels of health service can provide basic emergency obstetric care [30], however this is often not practicable for chronically short-staffed, resource-strained health facilities, especially for small services in remote, hard-to-reach areas. Five health facilities were involved in this study including government-run facilities and Catholic health services, which ranged significantly in size, funding, staffing, and services offered (Table 1).

**Table 1. Outline of health centres included in the study.**

| Health Service | Classification | Average No. of Births Annually | Maternity Workforce |
|---|---|---|---|
| Nonga General Hospital (Government) | Tertiary Hospital | 2160 | 31 (4 doctors, 17 midwives, 6 nurses, 4 CHWs) |
| St Mary's/ Vunapope Hospital (Catholic) | Rural Hospital | 2400 | 19 (2 doctors, 7 midwives, 2 nurses, 8 CHWs) |
| Kerevat Rural Hospital (Government) | Rural Hospital | 960 | 5 (2 midwives, 2 HEOs, 1 CHW) |
| Napapar Health Centre (Catholic) | Health Centre | 470 | 24 (2 midwives, 1 HEO, 9 nurses, 12 CHWs) |
| Malasait Health Post (Government) | Community Health Post | 0 | 2 (1 nurse, 1 CHW) |

HEO = Health Extension Officer, CHW = Community Health Worker

## Participants

Participants were all community members of East New Britain, PNG, and aged over 18. Women, partners, and healthcare professionals were recruited face-to-face from postnatal wards of participating health centres through convenience sampling. Healthcare managers and provincial health administrators (henceforth referred to as 'administrators') were recruited purposefully, to ensure views were captured from all levels of the provincial health system. All participants who were invited to take part were provided with written and verbal information and explanations about the study in Tok Pisin, Kuanua (common languages in East New Britain) and/or English. Participants were informed of the voluntary nature of the study with the ability to withdraw at any stage, and no participants declined the invitation to participate. Women and partners interviewed were not known previously to researchers, but some healthcare professionals, managers, and administrators were known to researchers from previous projects. Participants understood the researchers to be a part of the research team and local community, aiming to improve maternal healthcare in East New Britain.

## Data collection

The research team comprised four PNG national researchers; three female researchers (P.M., R.S., and P.H.), and one male researcher (D.K.). All four were experienced maternal and child health researchers, with qualifications in nursing, medicine, and public health; had extensive training in qualitative interview techniques; and had detailed knowledge of the East New Britain healthcare system, through both personal and professional experience. All members of the research team spoke English and Tok Pisin, whilst two of the researchers (P.M. and D.K.) also spoke Kuanua. Interview guides (S2 Table) were piloted by the research team in various health facilities in East New Britain to ensure appropriateness of questions asked and responses elicited. Interviews covered demographic details, experiences delivering or receiving maternal and newborn health care, perceptions of quality of care, men's involvement in care, and views around how care experiences could be improved.

Data were collected through semi-structured, in-depth interviews conducted from 1st September 2019 to 28th February 2020. Interviews lasted 30–60 minutes and were conducted in quiet, private locations in healthcare facilities by gender concordant facilitators as recommended by the study advisory team. During interviews, two researchers (one conducting the interview, and one taking field notes) were present in the room with participants. Interviews were audio-recorded, and field notes were cross checked with participants at the conclusion of the interview. There was no further follow up with participants. Interviews were conducted in Tok Pisin, Kuanua, or English (depending on participant preference), recorded, transcribed verbatim in the originally spoken language, then translated to English for analysis.

A two-day workshop on quality maternal and newborn health care, where a separate group of 35 community members, healthcare professionals, and healthcare managers attended, was held post data collection. This provided an opportunity to member-check and validate the data and preliminary findings. Whist data saturation was reached prior to the completion of data collection, the research team continued with the remaining scheduled interviews to ensure different stakeholders across all selected catchment areas were able to take part. This was especially important to understand varying quality maternal and newborn care definitions from different stakeholders, as was the aim of this project.

## Data analysis

Braun and Clarke's six-phase approach to thematic analysis was applied to analyse the data [31]. The use of thematic analysis allowed for interpretation of patterns of meaning in the data

**Table 2. Number and distribution of participants in the study.**

| Health Service | Women | Partners | Healthcare Professionals | Healthcare Managers | TOTAL | Administrators |
|---|---|---|---|---|---|---|
| Nonga | 5 | 2 | 2 | 2 | 11 | 4 |
| Vunapope | 4 | 2 | 2 | 2 | 10 | |
| Kerevat | 2 | 1 | 2 | 2 | 7 | |
| Napapar | 2 | 2 | 2 | 2 | 8 | |
| Malasait | - | - | 1 | 1 | 2 | |
| TOTALS | 13 | 7 | 9 | 9 | 42 | |

[32]. The computer software NVivo (Version 20, QST International Pty Ltd) was used to manage the data. Interview transcripts were translated to English, and the authors familiarised themselves with the data. A combination of an inductive and deductive approach to create initial codes was taken, with deductive codes identified through reviewing relevant literature. Codes were then synthesised into themes and sub-themes by three authors, which formed the coding tree, with associated supporting quotations documented, aided by Microsoft Word to organise the data. The authors then met to refine and agree on themes and sub-themes which emerged from the data. Preliminary results were presented to PNG national researchers to facilitate and ensure correct interpretation of findings, and suggest thematic and coding changes, which were integrated into the results. This was an opportunity for findings to be validated and greater context to be provided.

Exemplar quotes were chosen from coded data for inclusion in the results throughout the analysis process. Some quotes have been edited to facilitate readability. In the results, quotes are indicated in "*italics*". Where context and additional information has been provided, this is indicated with [square brackets]. Where non-relevant sections of text have been omitted, this is indicated with [. . .].

## Results

In total 42 stakeholders participated in the study, comprised of 13 women, seven partners, nine healthcare professionals, nine healthcare managers, and four provincial health administrators (Tables 2 and 3).

Four themes and 15 sub-themes were identified from the thematic analysis (Table 4). Theme results are presented in the order a client journeys through the maternity system; first

**Table 3. Participant characteristics.**

| Group | Total | Average Age [years] | Education (Completed Year 12) [% (n)] | Average Parity | Living with Partner [% (n)] | Attended ANC* [% (n)] | Facility Birth [%(n)] |
|---|---|---|---|---|---|---|---|
| Women | 13 | 25.8 | 46% (6) | 2.6 (range = 1–6) | 92% (12) | 92% (12) | 100% (13) |
| Partners | 7 | 30.0 | 0% (0) | - | - | - | - |

| Group | Total | Role (n) | | | | Average Years of Experience | |
|---|---|---|---|---|---|---|---|
| Professionals | 9 | Community Health Worker (4) Midwife (4) Nurse (1) | | | | 13.4 | |
| Managers | 9 | Midwife (4) Nurse (3) Health Extension Officer (2) | | | | 16.8 | |
| Administrators | 4 | Provincial Health Administrator (4) | | | | 19.0 | |

*ANC = Antenatal Care.

**Table 4. Themes and sub-themes.**

| Themes | Sub-themes |
|---|---|
| Theme I: Ensuring Access–Arriving at the Health Centre | Affordability |
| | Antenatal and intrapartum services |
| | Welcoming environment |
| Theme II: Experiencing Positive Care–What the Staff Do | Respectful care |
| | Timely care |
| | Companionship |
| | Communication |
| | Best practice |
| Theme III: Having the Bare Minimum–Resources Available to the Service | Physical resources |
| | Hygiene |
| | Guidelines and training |
| | Sufficient numbers of staff |
| | Privacy |
| Theme IV: Meeting Expectations–Outcomes of Care | Survival and safety |
| | Satisfaction |

accessing care and arriving at the healthcare centre, followed by the experience of care impacted by the resources available at the centre, and finally, the outcome of care.

## Theme I: Ensuring access–Arriving at the health centre

Women, partners, and healthcare professionals discussed the importance of affordability of care, stating that financial barriers can prevent women from accessing care, medications, and resources. Women appreciated low fees, with one woman saying "*The fees are low for antenatal and delivery. It's just five kina [kina is the PNG currency, roughly equivalent to $2.10 AUD] for clinic card and all other treatments are free*". However, women and partners described hidden fees in transport, drugs, and food. A woman from Vunapope said "*I will go to City Pharmacy and I will spend [. . .] more money again.*" Women may decide to not access care for financial reasons, as described by a healthcare professional: "*Sometimes they [women] won't have transport or money [so] they will just give birth in the village.*" Participants discussed how many births occur outside facilities, and that a substantial proportion of women do not access antenatal care. Staff described difficulty in delivering quality care to these women, as test results, due dates, and clinical history were unknown. Provision of antenatal care and facility-based births were specifically highlighted by a healthcare professional as an essential part of quality of care: "*One of the things is that we need to emphasise on mothers to attend antenatal clinics and give birth in health facility.*" A healthcare professional discussed that antenatal care improves outcomes; "*So if we give them that first care at their clinics at their pregnancy for those 9 months, we get a good outcome in the labour ward.*"

A positive and welcoming environment was an important aspect of quality of care. Being greeted kindly, and oriented to the hospital and ward were considered key components of quality of care by women, partners, professionals, managers, and administrators. A partner described the experience of being left outside of the health centre with his labouring wife, unable to gain access as the door was locked and unmonitored: "*[The staff] must be ready, they must come with the trolley and push the mother straight in. In here there was nothing when we arrived [. . .]. Every staff was busy doing their own things. So when my wife came she was standing outside calling in to the ward from outside.*" A healthcare manager discussed quality of care beginning with how staff members approach a patient when they enter the health centre.

"*Quality care means, not only with [having the] drugs, it's with our approach.*" The environment and interactions that women were initially exposed to, access to antenatal care and the affordability of care set the tone for the rest of the healthcare interaction.

### Theme II: Experiencing positive care–What the staff do

All participant groups discussed the importance of respectful care, including care which is holistic, and woman-centred. A healthcare manager summarised "*I'm looking at her holistically so meaning I have to look at her physiological state; her physical, her spiritual, emotional [state].*" Staff who had an open and kind approach, who spoke compassionately, and avoided negative language exemplified quality care according to participants. Some women described instances where they feared or experienced verbal and physical abuse (such as scolding or slapping) from healthcare workers. A healthcare manager described quality of care including a safe environment and trustworthy care; "*They [women] feel free to access [care] and they are happy and satisfied with the kind of care that they receive without any complaints, dissatisfaction, doubt or fear about of the staff and the facility.*" Additionally, participants discussed the importance of being provided with information, and autonomy to make treatment decisions. A healthcare professional emphasised the importance of autonomy; "*So guide them [women], and the choice is in their hand don't force them. . .it's their choice.*" A healthcare manager from the same facility agreed, stating "*We have to respect her rights.*" This included supporting women to choose their birthing position and seeking consent for procedures such as tubal ligation. The importance of clear communication was described, women wanted to be provided with information about procedures, as well as test and examination results. A health administrator stated that "*Quality maternal care mainly is a well-informed mother.*" A woman stated "*The doctors and nurses [. . .] must explain properly to the mothers who are coming to the hospital, [. . .] they [must] explain according to their level of their understanding.*" Women appreciated when healthcare professionals provided encouragement and support with kind language, especially in the absence of a companion, demonstrated by a woman who stated "*[We want staff who] will stay close and talk; 'you are like this and the baby is coming close now' or things like that, we want this kind of advice.*" Quality of care meant a relationship between the women and professional which put communication at the centre.

Timely care was a necessary component of quality of care by women, partners, and healthcare professionals. There were many situations described where women waited for extended periods of time before being seen and treated. Timely care was identified as an important domain of quality of care, as demonstrated by a partner who said: "*When the mother gives birth to the baby the mother and baby must be served quickly.*"

All groups discussed the importance of companionship for quality of care. Partners had mixed views, but most expressed a desire to be physically present during birth, so they could support their wife and in turn understand more about the birth process. A partner stated, "*It's good to be there during the time of delivery so that we can see and feel the pain that the mothers are going through so that we will take good care of them.*" However, partners, and companions more broadly, were sometimes barred from labour wards for several reasons including the lack of privacy for other labouring women. Some partners preferred not to be involved in birth due to cultural traditions. The benefits of companions in improving quality of care were noted by participants. For example, a healthcare manager stated, "*Since I do almost all the deliveries myself, I usually tell the husbands in assisting in things like making the beds for their wives.*" This was supported by a woman concurring "*If I go to the hospital with my guardian, [. . .] let my mother come in with me to the delivery room to stay [. . .] my mother can help me in some things.*" For many receivers and providers of care, companionship was a valued component of quality of care.

Best clinical practice was seen as a part of quality of care. Women and partners noted details about the sort of care they expected for their newborn, including thorough newborn checks and regular bathing, whilst healthcare professionals, managers, and administrators tended to discuss technical aspects, such as HIV and syphilis screening, complication management pathways, and preventing infection. A healthcare manager stated "*I see that quality is given [when the] baby is comfortable, the baby is happy, there is no infection. Sometimes, [. . .] the baby will develop a sepsis infection if we don't give quality care.*"

## Theme III: Having the Bare Minimum–What the service possesses

Utilities and resources available to health services was commonly discussed across participant groups. An administrator highlighted that a reliable water supply was a basic and essential component of quality care; "*Without the proper equipment [. . .] to me is not a quality care. And when there is no running water, there is no quality care too.*" Some women reported having vaginal exams completed under the light of a torch, as described by a woman; "*[Quality of care is] about the power black outs. There should be a standby generator set so in the event that there is power blackout, the generator can be switched on. It's not really good for them to use torch when they are checking [doing a vaginal examination on] us.*" Healthcare professionals described how facility infrastructure and layout could limit the delivery of a quality service. The lack of space restricted the number of beds in the labour ward, resulting in women sleeping on the floor, or being discharged early to make a bed available. A health professional stated, "*I think labour and postnatal ward needs extension because I see that mothers after childbirth, they stay in the postnatal ward and we discharge them less than the required days.*" Infrastructure limitations also contributed to privacy concerns of women and partners, as described by a woman; "*People walking in and out of the room [birthing room], they don't respect us the mothers who are sleeping and being cleaned up. Yesterday, I was really ashamed [because] of that guy [staff member] who was walking in and out of the room.*" Availability of medical equipment and drugs was emphasised by healthcare managers and professionals; a healthcare manager noted "*[For quality care], everything will be available in terms of equipment, medicine and staffing.*" Participants discussed a range of medications and equipment which aided good quality care, including hand sanitiser, paracetamol, misoprostol, amoxicillin, sterilisers, mosquito nets, and sanitary items. Healthcare workers described having to ration medication until the health service received its next allocation, to ensure it was available for a more urgent case that may arise. The provision of food and water was noted by women and partners as an important aspect of quality of care. A woman said, "*After the delivery of my baby and I am on the bed, they must help us with some food and drinks which we will gain our strength back*", and a partner from the same health service concurred; "*The health facility has to provide food.*" Often family members and companions were tasked with providing women with food and water, and in cases where there were no companions, women were left without.

General cleanliness and hygiene of wards, bathrooms, toilets, and showers was a component of quality of care discussed by women, healthcare professionals and managers. Women described blood stains on beds and sheets when they arrived at the health service, clogged sinks, used sanitary pads and rubbish covering the bathroom, and mosquitoes and ants throughout the ward. This did not align with how women defined quality of care. A woman said: "*Mothers' Modess [sanitary pads] are laying outside everywhere which is unhygienic, mothers' spit everywhere which is not good. [. . .] The sink was being blocked and the water is overflowing out of the bathroom to the pathway which people walk to the postnatal and labour rooms*", while another woman echoed the sentiment that bathrooms should be clean and hygienic, stating "*Inside the ward and delivery room in the shower must be cleaned at all times.*"

Ongoing staff training was recognised as fundamental to quality of care by healthcare professionals, healthcare managers, and administrators, demonstrated by a healthcare professional stating "*The skills that you have play a very important part, in that if you don't have skill [. . .] you won't give them [women] the best quality service.*" An administrator discussed the importance of ongoing training for healthcare professionals'; "*To do proper quality they [healthcare professionals] need to be taken out from their facilities and send them to attend short courses and trainings.*" All participant groups discussed having enough staff as an important pillar of quality of care, as described by a healthcare professional; "*Here, I feel that if there are enough staff working, then they [. . .] provide quality care.*" Being understaffed meant the desired standard of care could not be provided, especially in relation to respectful care, described by a healthcare manager; "*The staffing issue, we know that when we are very busy and we are limited, we are all humans so attitudes start coming in and that's how quality care will not be provided.*" Participants described that being understaffed led to delays in care and put more pressure on those at work, as they would have to work longer hours, and may miss breaks. In addition, being understaffed meant tasks like cleaning bathrooms and birthing environments were not completed as regularly as was needed, compromising hygiene. Healthcare professionals, managers, and administrators discussed the importance of having guidelines to follow to aid administration of high-quality care. An administrator stated, "*From the authority point of view I see that we need to control it [post-birth procedures for newborns] and we need to set some guidelines.*" They emphasised a reliance on guidelines to not only deliver quality care, but also to standardise practice throughout health services.

### Theme IV: Meeting expectations–Outcomes of care

All participant groups discussed the outcome of care, such as survival and a safe birth, as a very important part of quality of care, outlined by a healthcare professional stating "*In my opinion, I say that quality is the outcome.*" Emerging from the birth experience alive and well, with a healthy baby was the ultimate marker of quality for some participants, including this woman who stated "*They deliver my baby and gave me, I am happy and I am here. If I am alive, I am happy.*" Additionally, a birth free from life-threatening complications was hailed as quality care by women, demonstrated by a woman who said; "*I want the nurse to [. . .] help me until I deliver safe, delivered normal*", as well as a healthcare professional; "*The mother went into labour with-without any complications or without any problems; that's quality maternal care.*" Women, healthcare managers, and administrators emphasised women's satisfaction as important. A healthcare manager stated "*Quality is not in numbers. . . I'm looking at it as how this woman expresses as her feelings and satisfaction of the care that is been given and well. . . so that's quality maternal health.*" Administrators discussed staff also being satisfied with the care they provide as an important marker of quality of care, with one stating "*Would you [staff] think you have provided good work today? You go and sit down at the end of every afternoon and measure yourself, have you done quality, or you just go and roughly do the job just because you name was on for that day.*" From these perspectives, quality care was achieved when both providers and receivers of healthcare were satisfied with the birth experience.

Overall, the groups defined quality of care similarly, with all stakeholder groups discussing a welcoming environment, adequate staffing, respectful care, good outcomes of care, and best practice. A visualisation of the components of quality care as defined by each stakeholder group can be found in Fig 1.

## Discussion

This study aimed to define how different maternal and newborn care stakeholders in East New Britain, PNG, define quality of care. We found a significant overlap in the way that stakeholder

| Component of Quality of Care | | Women | Partners | Health Care Professionals | Health Care Managers | Provincial Health Administrators |
|---|---|---|---|---|---|---|
| **Theme I: Ensuring Access – Arriving at the Health Centre** | | | | | | |
| Affordability | | ✓ | ✓ | ✓ | | |
| Access to antenatal/intrapartum services | | ✓ | ✓ | ✓ | ✓ | ✓ |
| A welcoming environment | | ✓ | ✓ | ✓ | ✓ | ✓ |
| **Theme II: Experiencing Positive Care – What the Staff Do** | | | | | | |
| Respectful care | | ✓ | ✓ | ✓ | ✓ | ✓ |
| Timely care | | ✓ | ✓ | ✓ | | |
| Companionship | | ✓ | ✓ | ✓ | ✓ | ✓ |
| Communication | Clear/kind communication | ✓ | | ✓ | ✓ | ✓ |
| | Provision of information | ✓ | | ✓ | ✓ | ✓ |
| | Autonomy | ✓ | | ✓ | ✓ | ✓ |
| Best practice | | ✓ | ✓ | ✓ | ✓ | ✓ |
| **Theme III: Having the Bare Minimum – Resources Available to the Service** | | | | | | |
| Physical resources | Food and water | ✓ | | ✓ | ✓ | ✓ |
| | Adequate physical space | ✓ | ✓ | ✓ | ✓ | |
| | Medication | | | ✓ | ✓ | ✓ |
| | Medical equipment | | | ✓ | ✓ | ✓ |
| | Water and electricity | | | | | ✓ |
| Hygiene | | ✓ | | ✓ | ✓ | |
| Guidelines and training | | | | ✓ | ✓ | ✓ |
| Sufficient numbers of staff | | ✓ | ✓ | ✓ | ✓ | ✓ |
| Privacy | | ✓ | | ✓ | | ✓ |
| **Theme IV: Meeting Expectations – Outcomes of Care** | | | | | | |
| Survival and safety | | ✓ | ✓ | ✓ | ✓ | ✓ |
| Satisfaction | Women's satisfaction | ✓ | | | ✓ | ✓ |
| | Worker's satisfaction | | | ✓ | ✓ | ✓ |

**Fig 1. Summary of elements of quality of care as defined by each stakeholder group.** This diagram is a summary of the components of quality of care as defined by women, partners, healthcare professionals, managers, and administrators from East New-Britain, Papua New Guinea.

groups defined quality of care. A welcoming environment, having enough staff, best practice, good outcomes of care, companionship, and respectful care were domains of quality of care included by all stakeholder groups in their definitions. Autonomy, women's satisfaction with care, and access to reliable basic utilities such as water and electricity were other key components of quality care. Studies conducted in Tanzania, India, and Uganda also found that definitions between stakeholder groups had significant overlap [20–23]. One key difference in our study was that clinical and administrative provider groups focussed more on elements relating to provision of care and structural components of the healthcare system, while receivers of care (women and partners) focussed more on the care experience. Such differences between providers and receivers of care were seen in two studies conducted in Malawi [24, 25].

Our findings around 'Meeting expectations–outcomes of care' (Theme IV) highlights that some participants were primarily concerned with the outcome of care when considering quality; survival and safety, and avoiding complications. This is understandable, as comparably high rates of morbidity and mortality for mothers and babies in PNG mean that a significant proportion of women do experience complications [7, 8]. This does not take away from the other components of quality of care discussed in the other themes, as all components are important for a holistic, comprehensive, high-quality care experience. However, it does demonstrate the severity of the situation in PNG, and that adverse outcomes are common, and have profound impact on all stakeholders involved.

All stakeholders, but especially providers of care, identified that inadequate staffing created barriers to quality of care. These findings are consistent with other qualitative studies in Malawi, Uganda, and Tanzania [21, 22, 33]. In our study, healthcare professionals discussed

instances where they were physically unable to provide timely care, with women left waiting for substantial time periods. Staff found it harder to provide respectful care when physically and emotionally strained, as they were managing many women labouring with long hours and minimal breaks. Hygiene was also compromised, as there were insufficient cleaning staff to clean bathrooms and wards to create a hygienic and safe environment, which left healthcare professionals to fill this role, further increasing their workload. Previous research has outlined how inadequate staffing has negative implications for both healthcare professionals, and women, as staff experience increased workloads and perform tasks outside of scope, which can overburden health workers and result in loss of job satisfaction [33–35].

Allowing and encouraging companionship was a strong sub-theme highlighted by many participants, especially women and partners, and has also been found as an important aspect in defining quality of care in several qualitative studies across Nigeria, Malawi, Tanzania, and Afghanistan [20, 36–38], as well as PNG. Though barriers such as maintaining women's privacy on the ward, and cultural taboo were present, our study found companions were not only emotional and physical supports for women in labour but could assist with workforce shortage issues in completing tasks like delivering food, changing sheets and assisting with after birth hygiene and cleaning. Additionally, most male partners wanted to be involved in the birth process to better understand birth and what women go through, and to support their partners. A mixed methods study on companionship during labour in PNG echoes these findings, as well as noting companions could help with communication barriers, and even found healthcare professionals felt companionship helped labour progress [39].

## Strengths and limitations

To our knowledge, this project is the first to explore how a wide range of stakeholders define quality maternal and newborn care in PNG, and the Asia Pacific region more broadly. The exploratory, qualitative methodology helped to understand what is sought by stakeholders in maternal and newborn healthcare. Qualitative research allows groups who have not had the opportunity to participate in research previously, and traditionally been excluded from research, to have their voices amplified [27]. Many participants expressed they were grateful for the opportunity to participate in the research:

"*Just one thing to add; it's good you people came in to help with what we have discussed. For us mothers we cannot speak up for ourselves, it's difficult, we just listen and obey what they instruct us*"–Woman, Vunapope.

Another strength was the leadership of PNG national researchers from study inception through to completion, ensuring the project was relevant to the needs of, and informed by, the local community. To capture a diverse section of the community, this study was conducted with participants from five healthcare facilities, which varied in size and resources, and were based in urban, rural, and remote settings. Facilities included both government and faith-based facilities, and participants came from five diverse stakeholder groups. Findings from this study are therefore likely to be relevant to other LMIC settings where similar challenges to achieving quality care exist, especially throughout the wider Asia Pacific region.

A limitation of this study is that only women using a health service for labour and birth participated. Currently 37% of women giving birth in East New Britain, and 64% in PNG more widely, do not access a health facility for birth [9]. There are likely to be differences in how those who access facility care and those who do not define quality of care. was Almost half of women in our study, had completed 12 years of education, compared with an estimated 8% of women in the general population of PNG [40]. This may be reflective of the underlying population differences of who accesses a facility for maternity care in PNG. Despite this, we feel that

the experiences and views of these women are probably reflective of a wider group of women in PNG. Additionally, while medical professionals are an important stakeholder group, they were not included as participants in this study. This is because there are so few medical doctors in this region and they provide a minority of the maternity care.

An additional limitation is that this study had data collected prior to the COVID-19 pandemic. The impacts of COVID-19 on the health systems of many countries were severe, and may have had implications on quality of maternal and newborn care in PNG which have not been covered by this research.

## Implications

This study found there is a gap in how stakeholders in East New Britain, PNG define quality maternal and newborn healthcare and the care that is provided and received. Poor quality care has real world implications in how women seek maternal and newborn healthcare. In PNG, where the proportion of women birthing in a facility is already low, at 36% of all births [9], it is important women have positive experiences to ensure they return and encourage friends and family to utilise facilities for birth. Previous negative experiences and perceived poor quality of care at facilities impacts future utilisation of services and care seeking behaviours [41, 42].

Many changes required to improve maternal quality of care in PNG are not feasible in a short timeframe due to financial, workforce, and systemic constraints. In the short term, some recommendations that may be more feasibly implemented include provision of food and water by health facilities for mothers, encouraging companionship during labour and birth, and having dedicated cleaners to ensure the facilities are clean. This study's results showed companionship is an important part of quality of care definitions for key stakeholder groups and has previously been shown to be a low cost, effective quality improvement intervention in PNG [39]. Companionship is also supported as an important part of quality of care by healthcare professionals and women in Nigeria, Uganda, and Malawi [24, 36, 43, 44].

Longer term considerations could include an increase in the health workforce, (currently only around a quarter of the health workforce required to meet the needs of the population are available) [5], undisrupted access to utilities such as water and electricity, and infrastructure projects to design and build labour wards which ensure privacy and space for companions.

## Conclusion

Ensuring quality maternal and newborn healthcare is a complex challenge, especially in a low resource setting. To guide quality improvement initiatives that are maximally effective, culturally safe, and likely to be implemented, it is important to understand how stakeholders administering and receiving care define quality of care. This project has demonstrated key maternal and newborn healthcare stakeholders from East New Britain, PNG define quality of care as that which is respectful, welcoming, with facilities that are well-staffed, where best practice is employed, companions are welcomed, there is access to basic resources, and providers and receivers of care are all satisfied with the healthcare interaction. There is an opportunity to use these findings to inform the development of community informed and evidence-based quality improvement interventions in East New Britain, PNG, and across other areas of PNG and the Pacific where similar changes to quality of care are experienced.

## Supporting information

**S1 Checklist. Inclusivity in research questionnaire.**
(DOCX)

**S1 Text. Gender diversity statement.**
(DOCX)

**S1 Table. COREQ checklist.**
(DOCX)

**S2 Table. Interview guide.**
(DOCX)

## Acknowledgments

The Gutpela Sevis Study team includes (in alphabetical order) Delly Babona, James Beeson, Arthur Elijah, Priscah Hezeri, Stenard Hiasihri, Caroline SE Homer, Dukduk Kabiu, Angela Kelly-Hanku, Elissa Kennedy, Alison Morgan, Christopher Morgan, Pele Melepia, Michelle Scoullar, Naomi Spotswood, Rose Suruka, Lisa M Vallely, Joshua P Vogel, Pinip Wapi, Alyce N Wilson.

We would like to acknowledge the community of East New Britain, PNG for generously taking part in the study. We recognise and gratefully acknowledge the dedication and contribution by the HMHB study team in Kokopo and Melbourne, in addition to colleagues at PNG Institute for Medical Research. Our special thanks to the National Department of Health, the East New Britain Provincial Administration, the Provincial Health Authority, Catholic Health Services, and participating health facilities (Nonga General Hospital, St Mary's Vunapope, Kerevat Rural Hospital, Napapar Health Centre, Malasait Community Health Post) for enthusiastically facilitating our research team to work with them.

## Author Contributions

**Conceptualization:** Alyce N. Wilson, Pele Melepia, Delly Babona, Pinip Wapi, Michelle J. L. Scoullar, Caroline S. E. Homer.

**Data curation:** Alyce N. Wilson, Pele Melepia, Rose Suruka, Priscah Hezeri, Duk Duk Kabiu.

**Formal analysis:** Lachlan M. Faktor, Alyce N. Wilson, Pele Melepia, Delly Babona, Caroline S. E. Homer.

**Methodology:** Lachlan M. Faktor, Alyce N. Wilson, Pele Melepia, Delly Babona, Pinip Wapi, Rose Suruka, Priscah Hezeri, Duk Duk Kabiu, Lisa M. Vallely, Elissa Kennedy, Michelle J. L. Scoullar, Naomi Spotswood, Caroline S. E. Homer.

**Project administration:** Alyce N. Wilson, Pele Melepia, Caroline S. E. Homer.

**Supervision:** Alyce N. Wilson, Pele Melepia, Delly Babona, Caroline S. E. Homer.

**Writing – original draft:** Lachlan M. Faktor, Alyce N. Wilson, Pele Melepia, Delly Babona, Caroline S. E. Homer.

**Writing – review & editing:** Lachlan M. Faktor, Alyce N. Wilson, Pele Melepia, Delly Babona, Pinip Wapi, Rose Suruka, Priscah Hezeri, Duk Duk Kabiu, Lisa M. Vallely, Elissa Kennedy, Michelle J. L. Scoullar, Naomi Spotswood, Caroline S. E. Homer.

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
