## [Decision Letter · Decision Letter 0]

25 Jan 2024

PGPH-D-23-01862

“If I am alive, I am happy”: Defining quality of care from the perspectives of key maternal and newborn health stakeholders in Papua New Guinea

Dear Dr. Faktor,

Thank you for submitting your manuscript to PLOS Global Public Health. After careful consideration, we feel that it has merit but does not fully meet PLOS Global Public Health’s publication criteria as it currently stands. Therefore, we invite you to submit a revised version of the manuscript that addresses the points raised during the review process.

We look forward to receiving your revised manuscript.

Kind regards,

Hannah Tappis, DrPH, MPH

Academic Editor

Journal Requirements:

2. Please provide separate figure files in .tif or .eps format.

Additional Editor Comments (if provided):

Reviewers' comments:

Reviewer's Responses to Questions

**Comments to the Author**

1. Does this manuscript meet PLOS Global Public Health’s publication criteria? Is the manuscript technically sound, and do the data support the conclusions? The manuscript must describe methodologically and ethically rigorous research with conclusions that are appropriately drawn based on the data presented.

Reviewer #1: Yes

Reviewer #2: Yes

2. Has the statistical analysis been performed appropriately and rigorously?

Reviewer #1: N/A

Reviewer #2: N/A

3. Have the authors made all data underlying the findings in their manuscript fully available (please refer to the Data Availability Statement at the start of the manuscript PDF file)?

Reviewer #1: No

Reviewer #2: No

4. Is the manuscript presented in an intelligible fashion and written in standard English?

Reviewer #1: Yes

Reviewer #2: Yes

5. Review Comments to the Author

Reviewer #1: This article presents findings from a survey of stakeholders in maternal and newborn care in Papua New Guinea. A suitable number of postnatal women, partners, care providers and administrators from a range of facilities were interviewed. It is perhaps not surprising that health system users and providers have somewhat different orientations when it comes to defining “quality care”, and this finding is not very different from similar published work from sub-Saharan Africa, but this does appear to be the first study of its kind from the Western Pacific region.

Major comments

1. In many places it is hard to dissect out what are the desired standards of care which are not being met, from more general statements about what a facility should offer. While some quotes and researcher comments make it clear in places in others it is not. Of the participants, how many mentioned what would be considered gaps in quality of care?

2. Related to this the last line of the conclusion should be modified: “Participants believe that the current standard of care does not align with how they define quality of care”. There are examples throughout of individual participants highlighting shortcomings but no clear summary question that this refers to.

3. The authors bring in new topics in the discussion such as lack of cleaners and work burdens. These should be mentioned in the results.

4. Some of the statements in section 4 seem to rather work against all the previous sections in determining what is perceived as quality of care. This even refers to the “grab” in the title, which seems not to be the greatest choice to support the authors’ contention. Perhaps the authors could do more to address the nuances around this area- i.e. if the pregnancy outcome is good that is all that matters vs the numerous shortcomings outlined in the first three sections.

5. The authors acknowledge the lack of women who did not attend antenatal care in their sample, but this does limit the utility of their study.

Minor comments:

1. There are points in the results where the authors use the word “patient” inappropriately including to describe women who never attend antenatal care. Better to find another term whether it be “pregnant women” or something else.

2. Page 5 Line 3: which levels of facilities does this refer to, presumably not Health Posts?

3. Supporting information in the shape of interview questions are provided. Some are highlighted in yellow, others not. What is the difference? Were these data collected as part of a multi-component survey of which quality of care is just one part?

4. The timing of the study included the COVID pandemic. Were any allowances made for this in either study design or implementation, or in data interpretation? Staffing issues specific to the pandemic may have been operating.

Reviewer #2: Thank you for allowing me to review this manuscript. It is an important topic with real-world implications and actionable recommendations. It is overall well written.

I have several substantive recommendations, mainly around data analysis/interpretation/presentation, and several minor recommendations:

Pg. 4, paragraph 2, third to last sentence on page – I believe “10,0000” is meant to be “10,000” please double check if these measures are 10,000 or 100,000, but either way, clarify this number.

Pg. 6, last paragraph – interviews were conducted between September 1, 2019 and December 31, 2020. This, of course, was at the start of/during the COVID-19 pandemic. It makes me wonder what the COVID-19 context was in PNG at this time. What were the case counts? Were there policy restrictions? Were practices changed during this time? And, would any of these changes affect care or the answers provided for this study? It would be good to set the context to understand these since quality of care may have been different during the pandemic than at other times. For example, maybe there were restrictions affecting companionship, maybe women did not like wearing masks during their time in the hospital, maybe healthcare professionals were more overwhelmed than usual, or maybe cleaning staff were not going to work for fear of COVID-19 and affecting hygiene – I don’t know any of this since nothing is described about the effects or status quo during COVID-19. It does seem odd that COVID-19 is not mentioned at all in the paper by the authors or by the participants since care in general changed significantly during this time in many parts of the world.

Page 9 – Table 2 – recommend to list the health services in the same order as in Table 1

Page 9 – Table 3 – What is completed Year 12? Is this with regards to education? If so, I would add education. Also, if education, it surprises me that more women had a 12 year education than the partners (0%). Is this correct? I would find it surprising that women would be more educated than men in PNG. If so, does this reflect women who would likely deliver at a facility? Please clarify.

Page 10 Table 4 – I do not see “antenatal and intrapartum services” in the figure

Page 11, paragraph 2, sentence 3 quote – recommend to remove “-Partner” at end since already stated and it is the only quote with this descriptor. Recommend to either remove here or add descriptors to all quotes with sites.

Page 14 – mid-first paragraph – it sounds here that participants, including healthcare managers and professionals, are stating that sanitary items are important. Yet, in the figure, “hygiene” is only listed under women. Further, in the discussion, on page 17 (last paragraph, second to last sentence), you discuss hygiene from the perspective of professionals or “staff” as caring about hygiene as it increased their workload. I would re-consider whether “hygiene” is only cared about or mentioned from women as your evidence points to other stakeholders caring about hygiene.

Pg. 15, second paragraph, first sentence – I don’t see training in your subthemes or in the figure, yet, it seems to be important. It also surprised me that “guidelines” only fell under administrators and managers, as it would seem professionals would want to have guidelines and training to provide good care. Maybe guidelines here are describing facility policies only (I don’t think so)? If clinical practice guidelines, I would likely change “guidelines” to “Guidelines and training” and move it to the overlapped circle that includes administrators, managers and professionals, as long as professionals talked about standard of practice, which includes guidelines and training. These may also just fall under “best practice,” in which case, I would recommend to move “guidelines and training” to under “best practice” and explain these as part of best practice. Depends on what your data show how to organize this.

Pg. 17, second paragraph, sentence 3 – This sentence states that professionals were concerned about timely care, yet “timely care” is shown in the figure as only important to women and their partners. I would recommend to move “timely care” to the part of the diagram where women, partners and professionals overlap. This is also not discussed in the results section, but it is brought up in the discussion. Make sure not to introduce new concepts in the discussion section that have not been brought up in the results section. (e.g., hygiene and timely care). Double check that all concepts brought up by different stakeholders are reflected in the analysis and figure.

Pg. 19, last paragraph – from your analysis and description of the healthcare system, it would seem reasonable to add that short-term improvements may be made by addressing respectful care. Also, would it be feasible to recommend to hire non-medical personnel to help with hygiene concerns? It seems this is quite a concern to women and professionals and the description sounds dire and it would not require much training.

Figure 1 - Some suggestions have been made above regarding presentation of findings. In addition, it seems odd that professionals or partners do not care about “communication” as important to quality care. Please ensure this is in fact not mentioned by these groups, especially since “respectful care” (per all stakeholders) is described as entailing elements of communication.

“Privacy” is listed under women and partners, but on page 12, it is described that professionals or administrators (not clear) were also concerned about privacy, but this was manifested in other ways, such as excluding partners for concern of other patients’ privacy. This should still be included under the stakeholder group that had this concern.

It was not clear to me why best practice, welcoming environment, respectful care and adequate staffing were in the box, but other elements mentioned by all stakeholder groups were in the overlapping circles but outside the box (survival/safety/water/electricity/satisfaction with care/companionship). Please provide a clear reason for what was in or out of the box or rethink putting all these elements into the box.

Since the overall writing is well done, I will provide specific spelling / grammar recommendations:

Pg. 2 (second page 2), paragraph 2, sentence 2 – Add coma between “timely” and “equitable”

Pg. 3, paragraph 2, sentence 4 – change “publishd” to “published”

Pg. 6, paragraph 2, sentence 2 – Recommend to change to “All four were experienced maternal and child health researchers, with qualifications in nursing, medicine, and public health; had extensive training in qualitative interview techniques; and had detailed knowledge of the East New Britain healthcare system, though both personal and professional experience.”

Pg. 6, paragraph 2, sentence 4 – remove comma between “asked” and “and”

---

## [Decision Letter · Decision Letter 1]

24 Apr 2024

“If I am alive, I am happy”: Defining quality of care from the perspectives of key maternal and newborn health stakeholders in Papua New Guinea

PGPH-D-23-01862R1

Dear Mr Faktor,

We are pleased to inform you that your manuscript '“If I am alive, I am happy”: Defining quality of care from the perspectives of key maternal and newborn health stakeholders in Papua New Guinea' has been provisionally accepted for publication in PLOS Global Public Health.

Best regards,

Hannah Tappis, DrPH, MPH

Academic Editor

Reviewer Comments (if any, and for reference):

Reviewer's Responses to Questions

**Comments to the Author**

1. If the authors have adequately addressed your comments raised in a previous round of review and you feel that this manuscript is now acceptable for publication, you may indicate that here to bypass the “Comments to the Author” section, enter your conflict of interest statement in the “Confidential to Editor” section, and submit your "Accept" recommendation.

Reviewer #2: All comments have been addressed

2. Does this manuscript meet PLOS Global Public Health’s publication criteria? Is the manuscript technically sound, and do the data support the conclusions? The manuscript must describe methodologically and ethically rigorous research with conclusions that are appropriately drawn based on the data presented.

Reviewer #2: Yes

3. Has the statistical analysis been performed appropriately and rigorously?

Reviewer #2: N/A

4. Have the authors made all data underlying the findings in their manuscript fully available (please refer to the Data Availability Statement at the start of the manuscript PDF file)?

Reviewer #2: Yes

5. Is the manuscript presented in an intelligible fashion and written in standard English?

Reviewer #2: Yes

6. Review Comments to the Author

Reviewer #2: Thank you for making the revisions as suggested.

Other comments, Figure 1 - affordability is missing an "f", and there are f's also missing for "Sufficient number of staff" and under theme II - "What the staff do" - I am not sure if this is only in the copy I see or if this is indeed missing.

7. PLOS authors have the option to publish the peer review history of their article (what does this mean?). If published, this will include your full peer review and any attached files.

**Do you want your identity to be public for this peer review?** For information about this choice, including consent withdrawal, please see our Privacy Policy.

Reviewer #2: **Yes: **Dr. Elizabeth Alvarez
